# Comparison of Fine-Needle Biopsy (FNB) versus Fine-Needle Aspiration (FNA) Combined with Flow Cytometry in the Diagnosis of Deep-Seated Lymphoma

**DOI:** 10.3390/diagnostics13172777

**Published:** 2023-08-28

**Authors:** Yilei Yang, Bin Cheng, Dingkun Xiong, Dong Kuang, Haochen Cui, Si Xiong, Xia Mao, Yunlu Feng, Yuchong Zhao

**Affiliations:** 1The Division of Gastroenterology and Hepatology, Tongji Hospital, Tongji Medical College, Huazhong University of Science and Technology (HUST), Wuhan 430030, China; yangyileitj@163.com (Y.Y.); aruna0309@163.com (A.); haochencuitjmc@163.com (H.C.); xiongsi2015@126.com (S.X.); 2Department of Gastroenterology, Peking Union Medical College Hospital, Chinese Academy of Medical Sciences & Peking Union Medical College, Beijing 100730, China; xdk16@mails.tsinghua.edu.cn (D.X.); yunluf@163.com (Y.F.); 3The Division of Pathology, Tongji Hospital, Tongji Medical College, Huazhong University of Science and Technology, Wuhan 430030, China; kuangd@hust.edu.cn; 4Department of Hematology, Tongji Hospital, Tongji Medical College, Huazhong University of Science and Technology, Wuhan 430030, China; maoxia2009@163.com

**Keywords:** endoscopic ultrasound, tissue acquisition, fine-needle aspiration, fine-needle biopsy, lymphoma

## Abstract

Evidence comparing ultrasound endoscopy-guided fine-needle biopsy (EUS-FNB) with EUS-guided fine-needle aspiration (EUS-FNA) in deep-seated lymphoma tissue sampling is insufficient. This study aims to evaluate the diagnostic efficacy of immunohistochemistry (IHC) or flow cytometry (FCM) on specimens obtained from EUS-FNB and EUS-FNA in the diagnosis and staging of deep-seated lymphomas. This real-world, dual-center study prospectively evaluated all eligible specimens from patients who underwent EUS-FNB/FNA over an 8-year period. 53 patients were enrolled, with 23 patients in the EUS-FNB group and 30 patients in the EUS-FNA group. FNB yielded specimens with longer core tissues (0.80 mm [0.55, 1.00] vs. 0.45 mm [0.30, 0.50], *p* = 0.009) and higher scores of specimen adequacy [4 (3.75, 4.00) vs. 3 (1.00, 4.00), *p* = 0.025]. Overall analysis revealed that the diagnostic accuracy of IHC based on specimens acquired from EUS-FNB was significantly higher than that of EUS-FNA (91.30% vs. 60.00%, *p* = 0.013). After controlling confounding factors including lesion size and endoscopists, EUS-FNB with IHC maintained a higher-level diagnostic accuracy compared to EUS-FNA (OR = 1.292 [1.037–1.609], *p* = 0.023). When FCM was additionally used to analyze the specimen acquired from EUS-FNA, the diagnostic yield was significantly improved (ROC AUC: 0.733 vs. 0.550, *p* = 0.015), and the AUC of FNB alone or combined with FCM was 0.739 and 0.761. Conclusions: FNB needles generate higher histopathological diagnostic accuracy and specimen quality than FNA for the deep-seated lymphoma. Though the application of FCM significantly improves the diagnostic efficacy of EUS-FNA, FNB was still the preferred diagnostic modality with a shorter procedure time, comparable diagnostic accuracy, and better cost-effectiveness.

## 1. Introduction

With the increasing availability and advancements in diagnostic imaging tools, such as computed tomography (CT) and magnetic resonance imaging (MRI), the detection rate of mediastinal and abdominal lymphomas is on the rise [1,2,3]. A clear histopathological result is essential for accurate diagnosis and staging, which are crucial to the management primarily based on drug therapy. [4]. In contrast to superficial lymph nodes located in head, neck, and axillae regions, enlarged deep-seated lymph nodes are often inaccessible for percutaneous puncture biopsy of the target lesion tissue. Consequently, open exploration or the use of thoracoscopic and laparoscopic methods is frequently required. Although these procedures provide an ample amount of biopsy tissue for comprehensive pathological examinations, they are highly invasive and impose significant economic costs.

In the past decade, endoscopic ultrasound-guided fine-needle aspiration (EUS-FNA) has emerged as a widely used and effective tool for diagnosing thoracoabdominal deep-seated lymphoma, potentially replacing the aforementioned invasive methods [5,6,7,8,9]. However, the accuracy of EUS-FNA varies between 50.0% and 99.4%, depending on the location of the lymph nodes [10,11]. In recent years, novel fine-needle biopsies (FNB) with a novel end-cut design have gained widespread usage. These FNB needles theoretically offer better access to core tissue compared to conventional FNA needles [12,13].

Flow cytometry (FCM) combined with monoclonal immunofluorescence antibodies has emerged as a valuable technique in the field of medicine for the quantitative detection of cell surface, intracellular antigens, and membrane receptors [14,15,16,17,18,19,20]. Compared to immunohistochemistry (IHC), FCM offers several advantages, including rapid detection speed, high sensitivity, and cost-effective sample analysis. In cases where conventional histological evaluation is inconclusive, FCM can provide unique diagnostic insights. Recent studies have reported that the combination of FCM with EUS-FNA has yielded higher sensitivity and accuracy [21,22,23,24]. However, it is important to note that despite its potential benefits, due to the limited amount of tissue obtained, FCM cannot fully replace the histological information provided by biopsy specimens and may even result in misdiagnosis. Therefore, the National Comprehensive Cancer Network of the United States still recommends biopsy as the preferred diagnostic method and does not endorse fine-needle aspiration cytology examination [25].

Nevertheless, in the diagnosis of deep-seated lymph nodes, such as those located in the mediastinum and abdominal cavity, core needle biopsy or fine-needle aspiration biopsy combined with appropriate auxiliary techniques may be selected. To ensure a comprehensive examination, it is recommended to obtain multiple core tissue strips. The side oblique incision channel of the FNB puncture needle allows for the acquisition of complete tissue strips, theoretically yielding higher quality and more sufficient specimens for subsequent IHC and FCM analysis [26]. A retrospective study [27] was conducted to compare the sensitivity, specificity, and accuracy of EUS-FNB and EUS-FNA in lymph node sampling. The results revealed no significant difference in diagnostic sensitivity and accuracy between the two techniques. However, FNB demonstrated superior specificity compared to FNA. Despite this, for deep-seated lymphomas, there is currently a lack of comparative studies on IHC and FCM diagnostic accuracy of specimens acquired from EUS-FNB and EUS-FNA. Therefore, the objective of this study is to compare the adequacy and quality of specimens acquired from EUS-FNB and EUS-FNA and the diagnostic accuracy in the following analysis of IHC and FCM for deep-seated lymphoma.

## 2. Materials and Methods

### 2.1. Patients Enrollment

This present real-world, dual-center, single-blinded study included 53 patients with mediastinal and intra-abdominal lymphadenopathies, and finally diagnosed lymphoma. All patients underwent EUS-FNB/FNA and flow cytometry at Tongji Hospital, Tongji Medical College affiliated to Huazhong University of Science and Technology and Peking Union Medical College Hospital between April 2015 and June 2023. This study was approved by the institutional review boards of each participating center (TJ-IRB20220647) and registered at ClinicalTrial.gov (NCT05565066) accessed on 26 November 2022. Exclusion criteria for this study included patients with missing clinical and/or follow-up data; pregnancy, uncorrectable coagulation dysfunction (international normalized ratio > 1.5; platelet count < 50,000/mm^3^); history of anticoagulant medication within the last 1 week (e.g., aspirin, clopidogrel, warfarin); history of acute pancreatitis within the last 2 weeks, inability to tolerate ultrasound endoscopy (e.g., severe cardiopulmonary dysfunction); and those unable to provide informed consent for the procedure (e.g., individuals with psychiatric disorders).

### 2.2. EUS-FNB/FNA Procedure

Experienced endoscopists, who annually performed at least 150 EUS-FNB/FNA procedures, conducted the EUS-FNB/FNA. A curved linear-array echoendoscope (GF-UCT 260, Olympus, Tokyo, Japan) was used for the procedure. Before the procedure, all patients provided informed consent and received intravenous propofol anesthesia, with continuous electrocardiogram monitoring. EUS-guided tissue sampling was performed using either 22G, 25G FNA needles (EchoTip Ultra® from Cook, Shanghai, China; ExpectTM needle from Boston Scientific, Shanghai, China) or 20G, 22G FNB needles (EchoTip ProCore^®^ from Cook, AcquireTM from Boston Scientific) at the discretion of endoscopists.

During the procedure, endoscopists first performed a B-mode scan to identify the lesion, then the elastography and color Doppler system was utilized to determine the best path. Once the needle was inserted into the lesion, the stylet was removed and a 5–10-mL syringe was connected to the end of the needle applying suction to the needle’s lumen. Each pass adopted the “fanning” technique, the needle was moved back and forth swiftly within the lesion repeatedly for 20 times. After each pass, the needle is removed from the echoendoscope channel, and 0.1 mL of saline followed by 5 mL of air was injected into the needle cavity, to completely flush the aspirated specimen onto a smear slide. Blood on the slide was removed using a 1 mL syringe with a 27G needle, and the intact tissue strips were transferred into a 1.5 mL EP tube filled with 10% formalin for subsequent histopathological examination. Additional tissue strips were cryopreserved in a sputum cup filled with 5 mL RPMI-1640 for flow cytometry, while the remaining cell debris smears were sent for cytopathological examination. Additional passes were performed, at the discretion of endoscopist, until a sufficient amount of macroscopic tissue (white or yellow, maximum axis > 4 mm) was obtained, meeting the requirements of histological diagnosis. After the procedure, any observed complications were documented and properly treated. Patients were followed up by telephone from 4 to 7 days after the surgery to record any adverse events.

### 2.3. Immunohistochemistry

The aspirated samples from each pass were expelled onto separate slides with a stylet. Following this, 0.1 mL of sterile saline was flushed into the needle and followed with 5 mL of air. The macroscopically visible core tissue was transferred into Eppendorf tubes containing 10% formalin for histological examination and subsequently embedded in paraffin. Specimen sections were cut and stained with eosin and hematoxylin. (Sections of suspected AIP were further stained with IgG4, CD38, and CD138; sections of NET were stained with CgA, Syn, and CD56; sections of suspected mesenchymal tumor were stained with c-kit, CD34, DOG-1, α-SMA, Desmin, S-100; section of suspected lymphoma was stained with CD3, CD5, CD19, CD20, CD22, CD30, CD45RO CD79a, PAX5, BCL2.) Two pathologists (K. D. and C. X.), blinded to the type of needles used and clinical information, independently assessed all tissue samples obtained. When the 2 experts made a different diagnosis, the agreement was reached by consulting a third pathologist (D. Y.) and carefully discussing the findings (Appendix A).

### 2.4. Flow Cytometry

Specimens stored in RPMI-1640 were mechanically processed into single-cell suspensions and subjected to flow cytometry analysis. Based on the patient’s clinical manifestations, laboratory results, and other test results, the following mouse anti-human monoclonal immunofluorescent antibodies were used: CD2, CD3, CD4, CD5, CD8, CD10, CD11c, CD19, CD20, CD23, CD25, CD38, κ, λ, HLA-DR, CD30, CD45, CD103, CD138, Ki-67, bcl-2, CD16, CD56, CD57, CD94, CD161, CD158a/h, CD158b, CD158e, TCRαβ, TCRγδ, telomerase, perforin, etc. (Figure 1).

### 2.5. Specimen Evaluation

The tissue stored in EP tubes was embedded in paraffin and serially sliced. Initial hematoxylin-eosin staining (HE) was performed to evaluate the morphology and location of the tissue. Subsequently, IHC staining was conducted to confirm the diagnosis and classification. Additionally, other differential diagnostic markers and the prognostic marker p53 were also assessed based on the results of IHC. All cases were diagnosed according to the World Health Organization (WHO) criteria, considering morphology, immunohistochemistry, and/or molecular analysis.

The specimens were evaluated by two experienced pathologists (K.D. and D.Y.) specialized in hematology pathology. The two pathologists were blind to the type of needles used, the patient’s clinical information, and assessed all the tissue samples obtained independently. When the two experts made a different diagnosis, a consensus was reached after a careful discussion. The tissue integrity score was assigned to each section to assess specimen adequacy: 5 indicated sufficient material of high quality for accurate histological diagnosis (total material length > 1 × 10 high-power fields [HPF]); 4 indicated sufficient material of low quality for adequate histological diagnosis (total material length < 1 × 10 HPF); 3 indicated sufficient material for limited histological diagnosis; 2 indicated sufficient material for adequate cytological diagnosis; 1 indicated the presence of only small amounts of fat and normal tissue fragments; and 0 indicated insufficient tissue for diagnosis [28] (Figure 2).

### 2.6. Outcomes

The primary objective of this study was to compare the diagnostic yield of IHC and FCM based on FNA/FNB-obtained specimens for deep-seated lymphoma. The secondary objective was to compare the tissue quality, specifically the tissue strip length and tissue integrity score, between the FNB and FNA groups. The final diagnosis is typically established based on pathological reports of surgical, open biopsy specimens, or FNA/FNB histological evidence. In cases where no positive histological evidence was found, a benign lesion was considered if no progression or deterioration of the lesion was observed in the following imaging examinations and a follow-up period of at least 12 months.

### 2.7. Statistical Analysis

Continuous variables in this study were presented as the median (interquartile range) or the mean ± standard deviation (SD), while ranked data were expressed as the median and interquartile range (IQR). To assess the differences between the two groups, discrete variables were analyzed using the χ^2^ test or Fisher’s exact test, and continuous variables were analyzed using Student’s *t*-test or the Mann–Whitney U test. Multivariate stepwise logistic regression analysis was used to control for potential confounders, including needle size, lesion size, lesion site and endoscopists (R V.4.2.3). All data and statistical analyses were performed using SPSS version 26.0. A *p*-value of <0.05 was considered statistically significant.

## 3. Results

### 3.1. Baseline Data and Lesion Characteristics of Patients

From April 2015 to June 2023, a total of 53 eligible patients were enrolled, with 23 cases in the FNB group and 30 cases in the FNA group. The baseline data and lesion characteristics of the patients included in the statistical analysis are presented in Table 1. There were no significant differences between the two groups in terms of age, gender, lesion size, and lesion location, indicating comparability. The final diagnosis was determined by final surgical biopsy or EUS-guided tissue sampling results, including IHC and FCM. According to the final diagnosis, 51 out of the 53 patients were diagnosed with non-Hodgkin’s lymphoma. Among them, 29 cases were predominantly diffuse large B-cell lymphoma, while marginal zone lymphoma and follicular lymphoma were observed in 4 cases each. The subtype of B-cell lymphoma was unspecified in seven cases. Additionally, two cases were diagnosed with Hodgkin’s lymphoma. No adverse events related to EUS-FNB/FNA procedures were observed.

### 3.2. Characteristics of Needle and Sampling

In this study, various sizes of puncture needles were used, including 20 gauge, 22 gauge, and 25 gauge. The most commonly utilized needle size for EUS-FNB puncture was 20 gauge (78.26%), while for EUS-FNA puncture, it was 22 gauge (96.70%). To eliminate any confounding effects of various needle sizes on the study results, further multivariate regression analysis was conducted to compare the efficacy of FNB and FNA needles in terms of diagnosis. There was no significant difference observed in the number of passes required to obtain core tissue between FNB and FNA [4 [3, 5] vs. 4 [3, 6], *p* = 0.365]. However, the core tissue length of FNB-acquired specimens was significantly longer than that of FNA- acquired specimens (0.80 mm [0.55, 1.00] vs. 0.45 mm [0.30, 0.50], *p* = 0.009) and FNB-acquired specimens had significantly higher score of specimen adequacy than those obtained by FNA [4 (3.75, 4.00) vs. 3 (1.00, 4.00), *p* = 0.025], as shown in Table 2.

### 3.3. Diagnostic Accuracy

For all patients included in the analysis (*n* = 53), a final diagnosis was obtained through EUS-FNB or EUS-FNA histological findings, excisional biopsy, or surgery. The IHC diagnostic accuracy of FNB was significantly higher than that of FNA (91.30% vs. 60.00%, *p* = 0.013), as shown in Table 3. However, with regard to FCM, there was no statistical difference in the diagnostic accuracy between FNB and FNA (95.65% vs. 96.70%, *p* = 1.000) (Table 3). The diagnostic accuracy significantly improved in the FNA group when FCM was added compared to using IHC alone (ROC AUC: 0.733 vs. 0.550, *p* = 0.015). However, there was no statistically significant difference in diagnostic accuracy between FNB adding FCM and using IHC alone (ROC AUC: 0.761 vs. 0.739, *p* = 0.800) (Figure 3). When further stratified by lesion size, a significantly higher diagnostic rate in IHC was observed for lesions ≤ 20 mm using FNB compared to the FNA group (100.00% vs. 12.50%, *p* = 0.024) (Appendix A). For retroperitoneal lesions, subgroup analysis revealed that the IHC diagnostic rate of EUS-FNB was significantly higher than that of the EUS-FNA group, while there was no statistical difference between FNB and FNA group in FCM analysis (Appendix A).

### 3.4. Multivariate Logistic Regression

To evaluate the significant factors influencing the diagnostic accuracy of IHC alone and its combination with FCM, two methods were employed: univariate logistic regression and multivariate stepwise logistic regression. The univariate logistic regression analysis assessed the diagnostic accuracy of IHC alone as the dependent variable, considering five independent variables: needle type (FNA vs. FNB), needle size, lesion size, lesion site, and endoscopists. Two variables showed statistical significance: needle type (FNB vs. FNA) (OR: 7.000, 95% CI: 1.630–48.960, *p* = 0.019), needle size (OR: 0.384, 95% CI: 0.120–0.855, *p* = 0.042) and lesion size (OR: 8.750, 95% CI: 2.101–42.035, *p* = 0.004). Then, to determine the optimal set of variables for the diagnostic accuracy of IHC alone, stepwise logistic regression was conducted. The multivariate subset derived from the stepwise logistic regression model included needle type (FNB vs. FNA) (OR: 1.292, 95% CI: 1.037–1.609, *p* = 0.023) and lesion size (Table 4). Both methods indicated that FNB with IHC yielded a higher diagnostic accuracy compared to FNA. Additionally, the diagnostic accuracy of FNA and FNB were no longer statistically different when combined with FCM in both univariate logistic regression and multivariate stepwise logistic regression analyses (Appendix A).

## 4. Discussion

EUS-FNA has emerged as an important diagnostic tool for lesions in the gastrointestinal tract and its adjacent organs [3,29,30,31,32]. While fine-needle aspiration was previously heavily utilized for the diagnosis and staging of solid tumors, its diagnostic value for lymphoma has been the subject of debate [33]. The World Health Organization, the European Society of Medical Oncology, and the National Comprehensive Cancer Network recommend surgical resection as the preferred method for obtaining sufficient tissue for lymphoma diagnosis [1,25,34]. However, core needle biopsy or fine-needle aspiration biopsy, assisted with appropriate adjunct techniques, may be considered to aid in the diagnosis of deep-seated lymph nodes, such as those in the mediastinum and abdomen. Multiple core tissues are recommended to ensure a thorough histological examination. The lateral beveled orifice of the EUS-FNB needle allows for intact tissue, potentially yielding higher quality and more adequate tissue specimens for IHC and FCM. Aadam and colleagues evaluated both pancreatic (*n* = 73) and non-pancreatic (*n* = 67) lesions and concluded that FNB was superior to FNA in diagnostic yield when examining the non-pancreatic lesions (88.2% vs. 54.5%, *p* = 0.006) [35].A 2020 RCT statistically evaluated the diagnostic rate of EUS-FNB versus EUS-FNA for solid pancreatic lesions in the absence of ROSE [36], which showed that the diagnostic sensitivity of EUS-FNB (82%), accuracy (84%) were better than EUS-FNA (sensitivity: 71%, *p* = 0.03; accuracy: 51%, *p* = 0.03) [37]. Besides that, our previous study suggested that samples collected by FNB accurately identified 88.62% of all pancreatic lesions, whereas samples collected by FNA accurately identified 79.37% (*p* = 0.00468) in cytology analysis of with pancreatic masses [12]. However, there were no studies comparing the diagnostic efficacy of FNB versus FNA for lymphoma. Currently, flow cytometry testing has evolved as a crucial method for lymphoma diagnosis, playing a pivotal role in the diagnosis and differential diagnosis of B-cell non-Hodgkin’s lymphoma, T-cell non-Hodgkin’s lymphoma, and other malignant hematologic diseases, as well as aiding in tumor staging [15]. Therefore, this study was conducted to compare the diagnostic accuracy, specimen quality, and the number of passes between EUS-FNB and EUS-FNA, in conjunction with IHC or FCM, for deep-seated lymphoma lesions. This study demonstrates that FNB yielded specimens with longer core tissues (0.80 mm [0.55, 1.00] vs. 0.45 mm [0.30, 0.5], *p* = 0.009) and higher scores of specimen adequacy [4 (3.75, 4.00) vs. 3 (1.00, 4.00), *p* = 0.025]. Additionally, the FNB-obtained specimens show significantly better diagnostic accuracy than that of FNA in IHC. Specimens from EUS-FNA, combined with flow cytometry, resulted in an improved diagnostic rate (ROC AUC: 0.733 vs. 0.550 *p* = 0.015) and reached a comparable level with FNB.

To our knowledge, this current study represents one of the first attempts to compare the diagnostic efficacy of IHC and FCM based on specimens from FNB and FNA specifically in the diagnosis of deep-seated lymphoma. Previous studies comparing FNB and FNA needles predominantly relied on histopathological findings for the diagnosis of lymphoma, while FCM results were less frequently utilized. Even in large-scale systematic reviews and meta-analyses, FCM was not consistently employed in all cases of deep lymphoma. Furthermore, most of the randomized controlled trials (RCTs) included a broad range of lymph node lesions that underwent EUS, making lymphoma cases relatively rare, and FCM was not routinely employed [28,36,38]. In this study, the most common pathological diagnosis was diffuse large B-cell lymphoma, followed by follicular lymphoma and marginal zone lymphoma. Despite approximately 13% of patients presenting with an unspecified tumor subtype, based on histopathology and flow cytometry, these cases were classified as B-cell lymphoma, providing a foundation for determining appropriate clinical treatment strategies. The most frequent site of the lesions was the retroperitoneum, followed by the mediastinum, pancreas, and gastrointestinal tract. For FNB needles, the most commonly utilized needle diameter was 20 gauge, while 22 gauge was more frequently employed for EUS-FNA needles. This discrepancy in needle diameter composition partly explains the variations in diagnostic accuracy between the two groups. To enhance comparability between the groups and minimize potential baseline bias, a multifactorial logistic regression analysis was performed.

According to the latest WHO classification of lymphoma, the diagnosis of lymphoma necessitates a comprehensive assessment encompassing clinical features, morphology, immunophenotype, cytogenetics, and molecular pathology [4]. Consequently, there has been a growing emphasis on obtaining specimens through FNA or FNB needles in conjunction with appropriate pathological tests to facilitate further classification of lymphomas. Immunophenotypic analysis including IHC and FCM, enables the subclassification of lymphomas. In our study, the diagnostic accuracy of IHC from FNB was found to be significantly higher than that of the FNA group. This discrepancy could be attributed to the destruction of a significant amount of tissue collected during automated processing and sectioning. Such destruction can potentially result in misdiagnosis or missed diagnosis of lymphomas exhibiting mixed reactive lymphocytes or insignificant tumor cell heterogeneity. Compared to IHC, FCM offers the advantage of rapid, objective, quantitative, and multiparametric analysis. Although it cannot observe cytological morphology and in situ protein expression, FCM enables the analysis of individual cells using multiple antibodies. Our results demonstrate that EUS-FNA, with the aid of FCM, yielded better diagnostic rates over FNA-based IHC alone, which are as competitive as FNB. However, it should be noted that not all hospitals have adopted this as a routine clinical practice, possibly due to higher costs of FCM. The adoption of the FNB needle with IHC is much more easily implemented in clinical practice than a FCM program. Nevertheless, the presence of FCM can offer several advantages. Therefore, EUS-FNB is likely to be the preferred modality at most institutions because it does not require the presence of FCM to achieve excellent diagnostic accuracy, while being comparable in cost.

EUS-guided tissue sampling has been consistently reported as a reliable method for diagnosing lymph node lesions, and our findings further confirm it. Nevertheless, it is important to note that the size of the lesion may impact the diagnostic accuracy of FNA/FNB. Even after further stratification by lesion size, the success rate of IHC in FNB was significantly higher than that of FNA for lesions smaller than 20 mm. Moreover, there was no statistical difference between FNB and FNA when using immunohistochemistry or flow cytometry for the diagnosis of lesions larger than 20 mm. Therefore, it can be concluded that both FNA and FNB are suitable for larger lesions, whereas EUS-FNB exerts certain advantages for smaller lesions. Concerning the lesion sites, retroperitoneal lesions were the most commonly found locations, and the diagnostic rate of IHC in FNB was significantly higher than in the FNA group.

Despite being the first study comparing FNB versus FNA exclusively in lymphoma, we recognize some major limitations. Firstly, though specimens in this study were again prospectively evaluated uniformly, this is a real-world retrospective study with a lack of randomization and therefore inevitably subjected to selection bias and confounding factors. Additionally, multiple available needle sizes were used as it is a real-world study. In this study, ProCore accounted for a majority of the FNB needles. Given the limited use of other needles, we did not perform the comparative interclass analysis of different products. To eliminate these heterogeneities as possible, multivariate logistic regression was performed. A large randomized controlled trial is needed to confirm our findings.

## 5. Conclusions

FNB was superior to FNA for deep-seated lymphoma in IHC diagnosis. Though the combination of FCM and EUS-FNA significantly improved the diagnostic efficacy of FNA, which reached a comparable standard as FNB, we still recommend FNB as a preferred diagnostic modality with a shorter procedure time, better specimen adequacy, and cost-effectiveness.

## Figures and Tables

**Figure 1 diagnostics-13-02777-f001:**
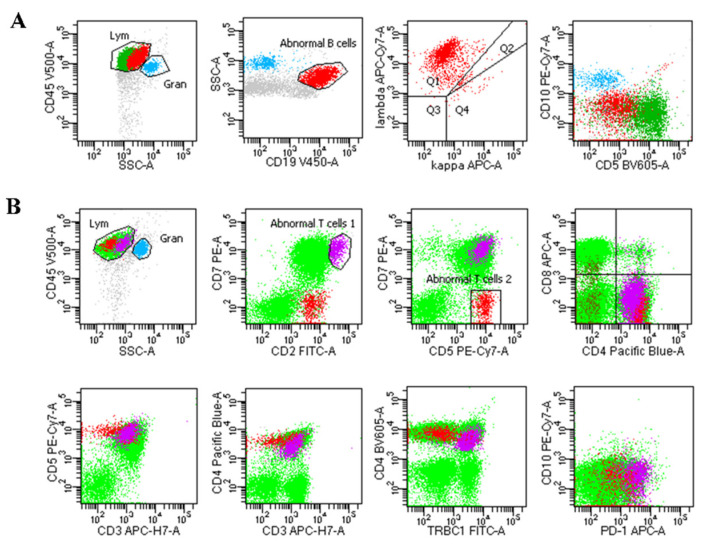
(**A**) A case of diffuse large B-cell lymphoma was diagnosed using EUS-FNB/FNA combined with FCM. The abnormal cells were shown in red. The abnormal cells expressed CD45, CD19, CD20, lambda while showing no expression of CD5 and CD10. (**B**) A case of T-cell lymphoma was diagnosed by EUS-FNB/FNA combined with FCM. Two abnormal T cell groups were observed, represented by purple and red. The purple group expressed CD45, CD3, CD2 (bright), CD5 (bright), CD4 (dim), PD1 (bright), TRBC1 (dim), CD7 while showing no expression of CD8 and CD10. The red group of T cells expressed CD45, CD2, CD5, CD4, PD1 while showing no expression of CD3, CD7, TRBC1 and CD10.

**Figure 2 diagnostics-13-02777-f002:**
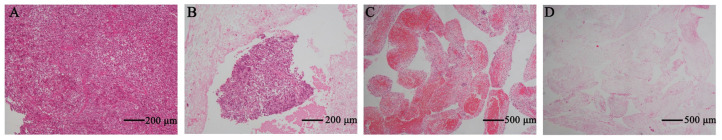
The tissue integrity assessments of specimens (H&E stained). Example of: (**A**) Score 5, sufficient material for adequate histological interpretation (core tissue length > 1 × 10 HPF, original magnification × 100); (**B**) Score 4, sufficient material for adequate histological interpretation (core tissue length < 1 × 10 HPF, original magnification × 100); (**C**) Score 3, sufficient material for limited histological interpretation (original magnification × 40); (**D**) Score 0, inadequate for diagnosis, based on previously reported system (original magnification × 40). Score 2 & 3 are measurements of cytological results, thus not exhibited here.

**Figure 3 diagnostics-13-02777-f003:**
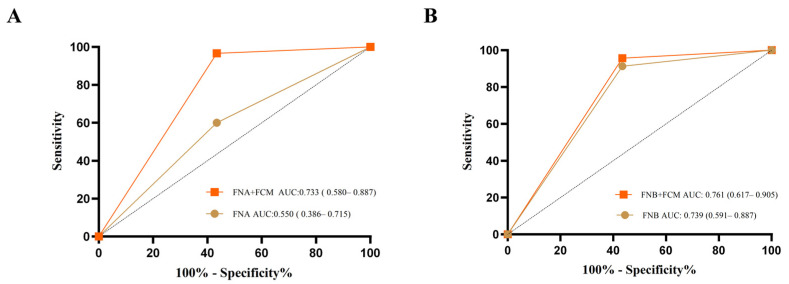
(**A**) ROC curves of EUS-FNA+ FCM vs. EUS- FNA. (**B**) ROC curves of EUS-FNB+ FCM vs. EUS- FNB. ROC curves: *X*-axis, 1-specificity; *Y*-axis, sensitivity. EUS-FNA, endoscopic ultrasound-guided fine-needle aspiration; EUS-FNB, ultrasound endoscopy-guided fine-needle biopsy; AUC, area under the curve.

**Table 1 diagnostics-13-02777-t001:** Baseline Patient and Lesion Characteristics.

Variables	FNB (*n* = 23)	FNA (*n* = 30)	*p*-Value
Age, years (SD)	54.70 (17.47)	51.87 (16.62)	0.479
Sex, male/female	12/11	13/17	0.359
Puncture site, *n* (%)			0.604
Mediastinum	1 (4.35%)	2 (6.70%)	
Retroperitoneum	20 (86.96%)	23 (76.7%)	
Pancreas	2 (8.70%)	3 (10.00%)	
GI tract	0 (0.00%)	2 (6.70%)	
Lesion size r, mm (SD)	38.12 (20.45)	30.90 (17.75)	0.176
Needle size, *n* (%)			<0.001 ***
20 gauge	18 (78.26%)	0 (0.00%)	
22 gauge	5 (21.74%)	29 (96.70%)	
25 gauge	0 (0.00%)	1 (2.00%)	
No. passes, median (quartile)	4 (3, 5)	4 (3, 6)	0.365
Subtype			1.000
NHL	22 (95.65%)	29 (96.70%)	
B cell lymphoma	20 (86.96%)	25 (83.40%)	
Diffuse large B-cell lymphoma	14 (60.87%)	15 (50.00%)	
Follicular lymphoma	2 (18.70%)	2 (6.70%)	
Mantle cell lymphoma	1 (4.35%)	0 (0.00%)	
Marginal zone lymphoma	1 (4.35%)	3 (10.00%)	
Unclassifiable	2 (8.70%)	5 (16.70%)	
T cell lymphoma	2 (8.70%)	4 (13.20%)	
Anaplastic large cell lymphoma	1 (4.35%)	1 (3.30%)	
γδT-cell lymphoma	0 (0.00%)	1 (3.30%)	
Small lymphocytic Lymphoma	0 (0.00%)	1 (3.30%)	
peripheral T cell lymphomas	1 (4.35%)	1 (3.30%)	
HL	1 (4.35%)	1 (3.30%)	

FNB, fine-needle biopsy; FNA, fine-needle aspiration; SD, standard deviation; GI, gastrointestinal; NHL, non-Hodgkin lymphoma; HL, Hodgkin lymphoma. *** *p* < 0.001.

**Table 2 diagnostics-13-02777-t002:** Specimen Quality.

	FNB (*n* = 23)	FNA (*n* = 30)	*p*-Value
Core tissue length, mm (quartile)	0.80 (0.55, 1.00)	0.45 (0.30, 0.50)	0.009 **
Score of specimen adequacy, median (quartile)	4 (3.75, 4.00)	3 (1.00, 4.00)	0.025 *

** *p* < 0.01; * *p* < 0.05.

**Table 3 diagnostics-13-02777-t003:** Comparison of diagnostic rate between FNB and FNA.

	FNB (*n* = 23)	FNA (*n* = 30)	OR (95% CI)	*p*-Value
No. of cases consistent with final diagnosis by IHC, *n* (%)	21 (91.30%)	18 (60.00%)	7.000 (1.380–35.511)	0.013 *
No. of cases consistent with final diagnosis by IHC+ FCM, *n* (%)	22 (95.65%)	29 (96.70%)	0.759 (0.045–12.812)	1.000

IHC, Immunohistochemistry; FCM, flow cytometry. * *p* < 0.05.

**Table 4 diagnostics-13-02777-t004:** Univariate and multivariate logistic regression analysis –diagnostic rate of IHC.

Variable	Univariate Logistic Regression			Multivariate Logistic Regression		
	Exp (b)	OR (95% CI)	*p*-Value	Exp (b)	OR (95% CI)	*p*-Value
Needle type (FNB vs. FNA)	7.000	1.630–48.960	0.019 *	1.292	1.037–1.609	0.023 *
Needle size	0.384	0.120–0.855	0.042 *	-	-	-
Lesion site	0.746	0.283–1.745	0.515	-	-	-
Lesion size	8.750	2.101–42.035	0.004 **	1.518	1.161–1.985	0.003 **
Endoscopists	2.917	0.821–12.133	0.112	-	-	-

* *p* < 0.05; ** *p* < 0.01.

## Data Availability

The datasets used and/or analyzed during the current study available from the corresponding author on reasonable request.

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
