# Peer review of "Comparison of Fine-Needle Biopsy (FNB) versus Fine-Needle Aspiration (FNA) Combined with Flow Cytometry in the Diagnosis of Deep-Seated Lymphoma"

_diagnostics, 2023, doi:10.3390/diagnostics13172777_

Round 1
Reviewer 1 Report
Yang et al. conducted a retrospective study to compare the diagnostic accuracy, specimen quality, and the number of passes between EUS-FNB and EUS-FNA, in conjunction with IHC or FCM, for deep-seated lymphoma lesions.
The topic is relevant and the study is well conducted. I have some concerns:
-Despite the low frequency of EUS-guided tissue acquisition of deep-seated lymphoma, could you specify how the sample size has been calculated?
-literature data about the superiority or not of FNB over FNA in other diagnoses should be mentioned in the discussion section;
-Please correct the typo in the 93-line (EUS-ENB)
Author Response
- We are deeply sorry for our negligence that this part did not describe in our manuscript. Though the specimens included in this study all underwent a prospective evaluation again, this is still a real-world retrospective study. We recruited all deep-seated lymphoma patients who underwent EUS-FNA/FNB at these two endoscopy centers during this time span. According to the advice, we specified our sample size based on the diagnostic accuracy in our study (FNB vs. FNA: 91.30% vs. 60.00%, a= 0.05, b= 0.2) and the sample size required is 26 in each group. And our sample size was 23 in FNB group and 30 in FNA group. Though with an inevitable bias, we carried out multivariate stepwise regression analysis to control the potential bias.
- We really appreciate this suggestion and have added this statement in Discussion.
- We are deeply sorry for this fault and have corrected it.
Reviewer 2 Report
While the article contributes to the ongoing discussion about the diagnostic methods for deep-seated lymphomas, it suffers from several limitations that compromise the strength and applicability of its findings. The lack of randomization, small sample size reduce the confidence in the study's conclusions. Additionally, article do not provide a seamless narrative, leaving the reader without a clear understanding of the study's significance and potential implications. Further research with larger, randomized controlled trials and more comprehensive discussions on the limitations and clinical relevance is needed to draw more robust conclusions in this area of study. Study include patient enrolled between 2015 to June 2022 only. Authors should add more data on patients enrolled since June 2022 and submit it for reconsideration.
Some areas of abstract and discussion lacks clarity in outlining the key findings and significance of the study. Authors can rewrite them to easy understanding with more patients data on hand.
Author Response
- We really appreciate this suggestion and have to admit that these limitations of our study. However, the application of EUS-FNA/FNB in the diagnosis of deep-seated lymphoma is relatively infrequent. We also included primary pancreatic lymphoma which is a highly rare disease. All these factors limit the sample size of our study. To make our findings more reliable, we carried out a multivariate stepwise regression analysis to control the potential bias. We still do realize that further randomized controlled trials with larger sample sizes are needed.
- In response, we enrolled all deep-seated lymphoma patients who underwent EUS-FNA/FNB with definitive diagnosis after June 2022 at the two endoscopy centers. And we further analyzed the data accordingly.
- We are deeply sorry for the ambiguity. We rewrote the abstract and discussion part and drew a clearer conclusion to make readers understand easier.
Round 2
Reviewer 2 Report
Thank you for the update. Added patients data shows that its rare disease and difficult to enroll high number of patients. Rewritten abstract looks good.